# Ultrasound-Guided Percutaneous Thermal Ablation of Renal Cancers—In Search for the Ideal Tumour

**DOI:** 10.3390/cancers15020518

**Published:** 2023-01-14

**Authors:** Milosz Jasinski, Marta Bielinska, Jerzy Siekiera, Krzysztof Kamecki, Maciej Salagierski

**Affiliations:** 1Department of Urology, Collegium Medicum, University of Zielona Góra, Zyty 28, 65-046 Zielona Góra, Poland; 2Department of Urology, Institute of Oncology, Romanowskiej 2, 85-796 Bydgoszcz, Poland; 3Department of Oncology, Institute of Oncology, Romanowskiej 2, 85-796 Bydgoszcz, Poland

**Keywords:** radiofrequency ablation, kidney cancer, ultrasound

## Abstract

**Simple Summary:**

Ultrasonography-guided percutaneous radiofrequency ablation is a relatively simple, safe and inexpensive treatment method for patients with small renal tumours and thus an attractive alternative in such cases. It has been, however, reported to be associated with an increased risk of recurrence in comparison with the current standard—partial nephrectomy. The aim of this study is to evaluate tumour characteristics associated with an increased risk of residual disease/recurrence and to find which tumours can be treated with ultrasonography-guided percutaneous ablation without an increased risk of recurrence. Ultrasonography-guided percutaneous radiofrequency ablation was safe and well-tolerated. Its effectiveness depended on tumour size, with best results for exophytic lesions smaller than 3 cm. Most of the recurrent or residual tumours were successfully re-treated with ultrasonography-guided percutaneous ablation.

**Abstract:**

Over the recent years, the progress in imaging techniques has led to an increased detection of kidney tumours, including small renal masses. While surgery is still the standard of care, there is a growing interest in minimally invasive methods. Ultrasound (US)-guided percutaneous ablation is particularly attractive because it is a safe and relatively simple procedure. In this study, we investigated the success of percutaneous radiofrequency ablation (RFA) in relation to kidney tumour diameter and location. Between August 2016 and September 2021, 253 patients with 259 renal tumours underwent US-guided RFA as a primary treatment in our institution. A total of 67 patients were excluded from this study. Abdominal computed tomography (CT) and tumour biopsy were performed before the procedure. Patients were followed with contrast-enhanced CT, the average follow-up time was 28 months. The studied group was composed of 186 patients with 191 renal tumours—only biopsy-confirmed renal cancers were included. During the follow-up, 46 cases of residual disease and 4 cases of local progression were found. There was a significant correlation between tumour size and the ablation success rate. The success rate was 73.5% and 87.6% for lesions ≤25 mm, 94.6% for lesions ≤25 mm and exophytic, 79.1% for lesions 26–30 mm and 84.4% for lesions 26–30 mm and exophytic, respectively. Four Clavien-Dindo grade ≥2 complications were observed. US-guided percutaneous RFA of T1a renal cancers is safe and well-tolerated. Its effectiveness depends on tumour size, with best results for exophytic lesions smaller than 3 cm. Most of the recurrent or residual tumours can be successfully re-treated with US-guided percutaneous RFA.

## 1. Introduction

In recent years, the progress in imaging techniques and wide introduction of ultrasonography (US) and computed tomography (CT) imaging has led to an increased detection of renal tumours, including small renal masses (SRM, kidney tumours smaller than 4 cm) [1,2]. While surgery is still the standard of care, there is a growing interest in minimally invasive treatment, such as thermal ablation [3]. The efficacy of thermal ablation has already been demonstrated, especially in case of SRMs, it is preferred for patients with renal masses ≤3 cm with contraindications or unwillingness to undergo surgery, especially people who are elderly, or have comorbidity, with bilateral tumours, and solitary kidney [4,5,6,7]. In addition to the good preservation of renal function, the percutaneous thermal ablation is also attractive because of much shorter hospital stay, lower cost, lower blood loss, shorter operative time, and acceptable morbidity in comparison to nephron sparing surgery (NSS) [2,4,8,9]. US-guided percutaneous ablation is particularly attractive because it is a relatively simple procedure and it can be performed by urologists skilled in US-guided techniques [2,10]. While thermal ablation and NSS have comparable rates of metastatic-free and disease-free survival, ablation has higher rates of local recurrence [2,3,10,11,12,13].

Over the years, several nephrometry scores, such as RENAL and PADUA, were developed in order to predict the risk of relapse and complications after treatment of renal tumours [3,14,15,16]. The RENAL score was also evaluated in patients undergoing renal tumour ablation [17]. However, both RENAL and its modification, mRENAL, were developed for surgical resections and its usefulness in context of percutaneous ablations remain controversial, as challenges of percutaneous ablation are different from those of surgery [3,14,18]. Recently, new scores—ABLATE and sABLATE—were developed by interventional radiologists with promising initial results [3,14].

In this study we investigated the primary success of radiofrequency thermal ablation (RFA), defined as no residual disease, local progression, or recurrence in diagnostic imaging, in relation to kidney tumour diameter and location.

## 2. Material and Methods

This retrospective observational study was approved by the institutional review board.

Between August 2016 and September 2021, 253 patients with 259 renal tumours underwent percutaneous RFA as initial treatment in our institution, recurrent lesions were not included in this study. This treatment was offered as an alternative to patients with T1 kidney tumours: those who are elderly and/or comorbid, unfit for surgery, or unwilling to undergo surgical resection, with tumour in single kidney.

Medical records were retrospectively reviewed for patients’ demographics, clinical data, and procedural details. Tumour anatomic features were evaluated in pre-procedural contrast-enhanced imaging (CT or MR). For each tumour size was measured and location in kidney was described as upper pole, central, or lower pole, lateral, medial-anterior, or medial-posterior. The tumour was described as exophytic (at least one-third exophytic) or non-exophytic (less than one-third). If a patient had more than one tumour, each one was evaluated separately.

All patients had contrast-enhanced imaging, either CT or MR, before the procedure. All tumours were biopsied, either during the ablation or before, as a separate procedure, patients without biopsy-confirmed RCC or with missing biopsy data were excluded from the study.

All thermal ablations were performed percutaneously, under US guidance. For each procedure, Covidien Cool-tip™ RF Ablation System was used. Ablation was performed with one probe, the length of ablation and eventual probe repositioning were decided according to size, shape, and characteristics of the lesion. As a standard, patients were discharged one day after the procedure.

Patients were followed by diagnostic imaging—contrast-enhanced CT or MR was performed at 3 months, 12 months after the procedure, then yearly. Follow-up scans were evaluated to assess the outcome. The treatment failure was the presence of enhancing tissue at the margins of the ablation volume in the first follow-up scan (residual disease) or within the ablation zone after at least one contrast-enhanced follow-up study demonstrating absence of viable tissue within the target tumour and surrounding ablation margin (local progression), as described in the literature [19]. Primary success was defined as complete eradication of tumour tissue, without residual disease, local progression nor local recurrence in later follow-up imaging. The follow-up time was calculated from the first ablation to the last diagnostic imaging available. Patients lost from follow-up or with lacking diagnostic imaging data were excluded from the study.

We performed statistical analysis using Statistica 8.0 software. Differences between variables were assessed using Mann–Whitney U-test. The χ-square test was employed to evaluate differences in qualitative variables. The Spearman correlation coefficient was used to evaluate the degree of association between quantitative variables. *p* < 0.05 was considered statistically significant.

## 3. Results

During the studied period, 259 renal tumours in 253 patients were treated with percutaneous RFA as initial treatment in our institution. A total of 67 patients with benign lesions, missing biopsy data, inconclusive biopsy, or lost from follow-up (no follow-up contrast-enhanced imaging available) were excluded from this study. Therefore, the studied group was composed of 186 patients with 191 renal tumours, 77 (41.4%) were females, and 109 (58.6%) males. Mean and median age of the population was 66.9 and 68 years, respectively. Mean and median tumour diameter were 26.5 and 25 mm. Most of the tumours were exophytic (*n* = 125, 65.4%), they were more often located in the central part of the kidney (*n* = 105, 55.0%), and more often lateral (*n* = 135, 70.7%). There were 23 tumours in a single kidney. There were 2 bilateral kidney tumours, one patient with 2 tumours in one kidney and one had 3 tumours in one kidney.

The procedure was generally well tolerated, we registered four Clavien-Dindo grade ≥II complications: one grade II bowel injury treated conservatively, one grade III bowel injury required surgical intervention, one grade III retroperitoneal haemorrhage requiring nephrectomy, and one case of procedure-related death because of septic shock. In addition, we registered five cases of Clavien-Dindo grade I complications—prolonged postoperative pain or fever requiring one day of additional observation.

Mean follow-up time was 28 months, and median was 23 months (range 3–74 months).

During follow-up, 46 cases of residual disease were found (enhancement in CT/MR 3 months after the procedure). In four cases local progression was found in follow-up later than after 3 months (in two cases 1 year, in one case 2 years, and in one case 3 years after initial procedure), despite initially complete ablation (no enhancement in CT/MR 3 months after procedure). These cases were treated with repeated thermal ablation—25 cases (one additional procedure in 21 cases, two procedures in 2 cases and three procedures in 2 cases)—one of them was microwave ablation, NSS–2 cases, nephrectomy–5 cases (two of them were nephrectomy after failed repeated thermal ablation), selective embolization and thermal ablation—8 cases (one additional procedure in 3 cases, two procedures in 5 cases and three procedures in 1 case), selective embolization—3 cases and observation—7 cases. A total of 36 of these patients were recurrence-free after this additional treatment and 14 with persistent disease. For these 36 patients, the mean recurrence-free time, calculated from the last procedure to the last follow-up imaging available was 24 months, and the median was 24 months (range 3–60 months). Patients after repeated ablation was followed according to the same scheme, contrast-enhanced CT or MR at 3 and 12 months, respectively, then yearly. The average time form initial treatment to the start of retreatment in cases of residual disease was 4 months.

There was a significant difference between tumours with and without primary ablation success in diameter and lateral location (Table 1).

The primary ablation success rate was 72.7% and 87.6% for lesions ≤25 mm, 94.6% for lesions ≤25 mm and exophytic, 79.1% for lesions 26–30 mm, and 84.4% for lesions 26–30 mm and exophytic, respectively—the details are presented in Table 2.

There was a specific subgroup of 29 patients—the ones without any significant comorbidities and functioning contralateral kidney. There were two cases of residual disease and one local progression in this group—the primary ablation success rate was 89.6% (vs. 71.6% in the rest of patients, *p* = 0.04). These patients were significantly younger than the rest, had smaller lesions, and more lateral lesions (Table 3). There was one case of Clavien-Dindo grade II complication in this subgroup.

## 4. Discussion

In recent years, thermal ablation has been gaining popularity, and is now accepted as one of the treatment methods in SRMs, including successful treatment of obese patients and cystic lesions [3,20,21,22,23]. Some authors reported comparable results of thermal ablation and NSS in selected patients [9,20,24].

It must be stressed that percutaneous RFA is a minimally invasive treatment and can in certain cases be performed under local anaesthesia [8,13]. It is associated with shorter hospital stay than NSS (even robotic) and low complication rate, even in more challenging scenarios, such as endophytic tumour or tumour in single kidney [9,25,26].

NSS is a well-established, well-known, thoroughly studied and described treatment method. It has become both the golden standard and the treatment of choice in T1a renal tumours. Due to its popularity, the treatment method itself and its indications are known to most of urologists. Scores, such as RENAL and PADUA, have been developed to help in qualification to NSS [15,16]. They have been established and validated based on large groups of patients. In contrast, percutaneous thermal ablation of renal masses, even the US-guided one, is much less popular, especially among urologists. There are also much less large, high-quality studies on thermal ablation of renal masses than on NSS.

It is an obvious fact that not all small renal masses are equal, both from surgical and percutaneous ablation point of view. However, while there are scores to help in qualification to surgery, the qualification to percutaneous ablation is based mostly on surgeon’s individual experience. Due to significant differences between NSS and percutaneous ablation, the surgical scores are unreliable in prediction of difficulties encountered in percutaneous ablative techniques [3,14,18]. Therefore, ABLATE, a specific score, was developed for the percutaneous ablation of renal masses.

Both ABLATE and its modification sABLATE were developed based on groups of patients treated with cryoablation or microwave ablation, mostly under CT guidance. There are some potential differences between US- and CT-guided ablation in the influence of tumour localization. For example, upper pole tumours may be more challenging for US-guided ablation due to the presence of ribs, while probe inclination from axial route may increase the difficulty of CT-guided ablation, but not the US-guided ablation. Therefore, while being useful, the scores developed for CT-guided ablation may not directly translate into US-guided ones.

In our study we found that the size of the tumour was the most important factor correlating with the rate of primary technical success of RFA. Others have also reported significantly worse results of RFA in tumours above 30 mm [27]. For the smallest (diameter ≤25 mm), exophytic lesions the primary success rate was almost 95%. Interestingly, the location in upper pole was not a significant factor for the lesions ≤25 mm. For the lesions 26–30 mm the situation is somewhat different. If we exclude lesions located in upper pole, the primary success rate in this group was 87% and almost 92% for exophytic lesions. This may be the answer to the question ‘what is the perfect kidney tumour for the percutaneous, US-guided thermal ablation?’—it is an exophytic lesion, smaller than 25 mm or smaller than 30 mm, and located in the central part or lower pole of the kidney.

The correlation between tumour size and the primary ablation success has also another aspect. While this rate is good for smallest lesion and still good for some 25–30 mm, it rapidly decreases with the size of the tumour above 30 mm. This may be an argument for more careful qualification of patients with SRMs, especially those with lesion diameter around 25 mm or more, to surveillance with diagnostic imaging alone, without biopsy. A significant proportion of these lesions is malignant and their growth may in some cases mean reducing the chance for highly effective minimally invasive treatment [28,29].

There is a specific subgroup of patients, the younger ones, without any significant comorbidities and competent contralateral kidney. These patients would be good candidates for NSS. The majority of lesions in this subgroup were ‘perfect tumours’, as described above.

The primary ablation success rate in this group was almost 90% and the three cases of residual disease/recurrence were successfully treated with repeated ablation. This good oncological effect, together with low risk of complications (one grade II) and short hospital stay, make percutaneous ablation a viable option in treatment of such patients. Moreover, we have already investigated the use of RFA in similar patients [10]. However, the success rate from current study is significantly higher than the 74% from the previous one. This may be due to the progress in ultrasonographic equipment as the first ablation from the current study was performed over six years after the last ablation from the previous study.

Percutaneous ablation is the treatment offered mainly to frail and/or comorbid, elderly patients, but indications for such treatment may also be extended to some other patients with small renal masses [20]. The reasons to choose this method are low morbidity rate and safety for the patient, despite the possible and reported in some studies higher risk of recurrence [12,29,30]. This was confirmed by our observations as the complication rate was relatively low, despite including many elderly patients and comorbid patients. However, there are two groups of factors that should be considered when selecting patients for percutaneous ablation: the patient-related ones and the tumour-related ones. The first one includes age, comorbidities, shorter life expectancy, solitary kidney, or kidney insufficiency, unwillingness to undergo surgery, and can generally be described as ‘poor candidates for surgery’. The second one includes tumour characteristics, such as size and location, and can be described as ‘good candidates for ablation’. These patients may still benefit from percutaneous ablation even if they are suitable for surgery, due to the comparable oncological effect and lower morbidity.

It can be expected that the results of US-guided, percutaneous RFA should be better in patients with preferable tumour characteristics than in those with only patient-related indications. Indeed, in our study, there patients from the subgroup without comorbidities, most of whom were offered percutaneous RFA based on preferable tumour characteristics, had higher primary success rate than the rest, who were often selected mainly because of being poor candidates for surgery.

This study confirms that percutaneous RFA is safe and well-tolerated in all patients groups, including elderly and comorbid ones—the complication rate was lower than the ones reported for NSS [31,32,33,34]. US-guided, percutaneous RFA can be an effective treatment method for certain patients with T1a renal cancers, with residual disease/recurrence rate lower than 10% during a mean follow-up time of over 2 years in selected patients, achieved with only one RFA session, with short hospital stay, without general anaesthesia and with relatively low morbidity. With appropriate, careful follow-up, most of these residual disease/recurrence cases can be identified and successfully re-treated, in majority of cases with minimally invasive techniques. This study was focused on primary ablation success. When re-treatment of residual disease and recurrences is included, the total disease-free rate would be 91.6%, which is comparable to results reported in other studies [27,35,36]. Slightly better results in our group should probably be attributed to shorter follow-up or including less T1b tumours.

This study has some interesting aspects. First, we analysed only patients with biopsy confirmed RCC, patients with benign tumours or missing data were excluded. This is important as many other studies also include patients with benign, unknown, or inconclusive biopsy results. Second, this study only included patients treated in a rather uniform way—only percutaneous, US-guides, or radiofrequency ablation. There were no microwave ablations (except one repeated ablation), no cryoablations, and no CT-guided ablations included in this study. In addition, all the procedures (and the vast majority of pre-procedural biopsies) were performed by two operators: M.J. and J.S., who are urologists with experience in US-guided procedures.

This study has several limitations. The retrospective nature has probably caused selection bias. The tumours with medial location, especially medial anterior, were considered more difficult for US-guided percutaneous RFA. Such patients were less likely to be treated with this method, especially if they were good candidates for NSS. A relatively large group of patients was also excluded from the study, either due to missing data or lost from follow-up. The median follow-up time was 23 months, but some patients were followed for only 3 months. On the other hand, the investigation of long-term results of RFA was not the primary aim of this study as most residual disease/recurrences are detected within the first 2 years [27,37].

## 5. Conclusions

In conclusion, US-guided percutaneous RFA of T1a renal cancers is safe and well-tolerated. Its effectiveness depends on tumour size, with best results for exophytic lesions smaller than 3 cm. Most of the recurrent or residual tumours can be successfully re-treated with US-guided percutaneous RFA. It is probable that indications to US-guided percutaneous RFA could be expanded and possibly this method could be the treatment of choice for some small renal masses not only in comorbid patients and elderly patients, but further research is required.

## Figures and Tables

**Figure 1 cancers-15-00518-f001:**
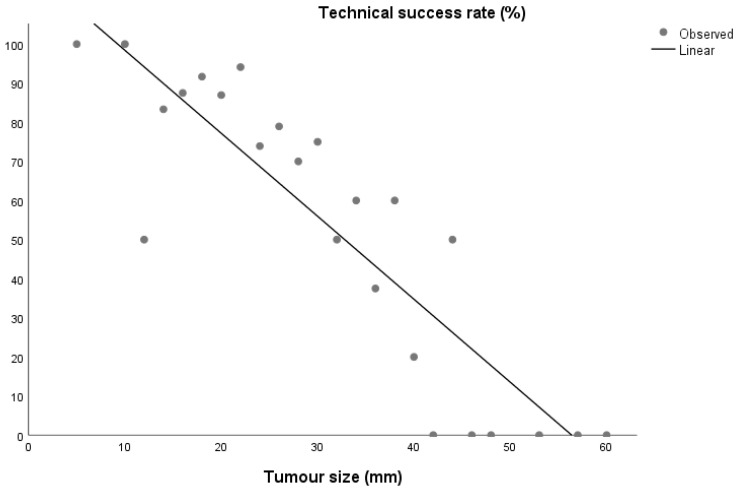
Primary ablation success rate according to tumour size.

**Table 1 cancers-15-00518-t001:** Characteristics of the ablated tumours.

	Primary Success	No Primary Success	*p*
*n*	139	52	
Age (mean ± SD) [y]	66.1 ± 10.6	70.1 ± 9.6	*p* = 0.014
Diameter (mean ± SD) [mm]	24.0 ± 6.7	33.0 ± 10.0	*p* < 0.001
Diameter (%):			*p* < 0.001
≤25 mm	60.8	23.1	
25–30 mm	23.0	17.3	
30–40 mm	13.7	40.4	
>40 mm	0.7	19.2	
Exophytic (%)	68.8	61.5	NS
Location (%):			NS
Upper pole	20.9	25.0	
Central	54.7	57.7	
Lower pole	24.4	17.3	
Laterality (%):			*p* = 0.023
Lateral	76.2	59.6	
Medial posterior	11.5	23.1	
Medial anterior	12.3	17.3	

There was a significant correlation between tumour size and the primary success rate (Spearman r = −0.886, *p* < 0.001) [Figure 1].

**Table 2 cancers-15-00518-t002:** Primary ablation success according to tumour size and location.

	Lateral		Medial Posterior		Medial Anterior	
	*n*	No Primary Success	*n*	No Primary Success	*n*	No Primary Success
Upper pole	30	6 (20%) *	10	6 (60%)	2	1 (50%)
Central	75	20 (27%) *	13	4 (31%)	18	6 (33%)
Lower pole	32	5 (16%)	5	2 (40%)	6	2 (33%)
Diameter ≤ 25 mm						
Upper pole	14	1 (7%)	2	0		
Central	43	7 (16%)	4	1 (20%) *	7	1 (14%) *
Lower pole	21	2 (10%)	3	0	3	0
Diameter ≤ 25 mm, exophytic						
Upper pole	8	0	2	0		
Central	24	3 (12%)	3	0	3	0
Lower pole	13	0	3	0		
Diameter 26–30 mm						
Upper pole	6	1 (17%)	6	4 (67%)		
Central	12	1 (8%)	4	0	6	2 (33%)
Lower pole	7	1 (14%)			2	0
Diameter 26–30 mm, exophytic						
Upper pole	4	1 (25%)	4	2 (50%)		
Central	9	0	3	0	4	1 (25%)
Lower pole	6	1 (17%)			2	0

* Includes one case of local progression.

**Table 3 cancers-15-00518-t003:** Characteristics of ablated tumours in patients with and without comorbidities.

	No Comorbidities	With Comorbidities	*p*
*n*	29	159	
Age (mean ± SD) [y]	52.7 ± 10.7	69.5 ± 7.8	*p* < 0.001
Diameter (mean ± SD) [mm]	23.1 ± 5.0	27.2 ± 9.3	*p* = 0.02
Diameter (%)			*p* = 0.02
≤25 mm	68.9	44.7	
26–30 mm	27.5	22.0	
30–40 mm	3.6	25.8	
>40 mm		7.5	
Exophytic (%)	62.1	68.5	NS
Location (%):			NS
Upper pole	24.1	19.5	
Central	41.4	59.1	
Lower pole	34.5	21.4	
Laterality (%):			*p* = 0.04
Lateral	86.2	67.2	
Medial posterior	6.9	16.4	
Medial anterior	6.9	16.4	

## Data Availability

Data generated or analysed during this study are included in this article. More detailed data are not publicly available due to their containing information that could compromise the privacy of patients. Further enquiries can be directed to the corresponding author.

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
