# Peer review of "Ultrasound-Guided Percutaneous Thermal Ablation of Renal Cancers—In Search for the Ideal Tumour"

_cancers, 2023, doi:10.3390/cancers15020518_

Round 1
Reviewer 1 Report
There are some minor errors in need of grammar/syntaxis correction.
Materials and methods:
- Was there a limit tumor size (maximum) to exclude a patient from the study?
- Could you better describe what were the criteria for offering a treatment with ablation, rather than with NSS in those patients?
Results:
- What would the final medium follow up time be? It should be added to the results of the study.
- Grade I and II Clavien Dindo complications should be also mentioned.
- There should be a clear definition between complete and incomplete treatment-ablation, early versus late recurrence and finally what you consider to be treatment failure.
- A clear definition between failure and incomplete resection is also useful as the word incomplete might refer to an initial technical aspect while failure has to do with the follow up.
- On table 2 early and late recurrences should be added
- From line 97 to 98 it would be helpful to better define/describe the number of tumors per kidney and the number of kidneys involved.
- From line 102 to 103 what was the thought behind excluding the patients? Aren't embolisation and death supposed to be included as "failures"?
- From line 106 to107 it would be nice to indicate when exactly the recurrence in those 4 cases was found
- From line 114 to 115 it would be helpful to provide the exact time frame of additional treatments and the recurrence free time of reference (starting from initial resection or maybe from the last intervention).
- From line 133 to 134 a definition of late recurrence would be needed. (Is it later than the second follow up at one year?)
Discussion:
- In line 147 are there any references citing that RFA offers shorter hospital stay than RARN and low complication rates?
- It would be helpful to provide a comparison of your results to the results of other studies concerning RF ablation such as:
1. Abu-Ghanem, Y., et al. Limitations of Available Studies Prevent Reliable Comparison Between Tumour Ablation and Partial Nephrectomy for Patients with Localised Renal Masses: A Systematic Review from the European Association of Urology Renal Cell Cancer Guideline Panel. Eur Urol Oncol, 2020. 3: 433
2. Psutka, S.P., et al. Long-term oncologic outcomes after radiofrequency ablation for T1 renal cell carcinoma. Eur Urol, 2013. 63: 486.
3. Johnson, B.A., et al. Ten-Year Outcomes of Renal Tumor Radio Frequency Ablation. J Urol, 2019. 201: 251.
4. Chang, X., et al. Radio frequency ablation versus partial nephrectomy for clinical T1b renal cell carcinoma: long-term clinical and oncologic outcomes. J Urol, 2015. 193: 430.
Reviewer 2 Report
Dear Editor
In this article, the authors discussed the progress in imaging techniques for kidney tumors, including small renal masses. The authors declare that ultrasound (US) guided percutaneous ablation is particularly attractive because it is a safe and relatively simple procedure. In this study, authors investigated the success of thermal ablation in relation to kidney tumor diameter and location. Further, Abdominal computed tomography (CT) and tumor biopsy were performed before the procedure. Additionally, the patients were followed with contrast-enhanced CT, the average follow-up time was 28 months. The studied group comprised 186 patients with 191 renal 17 tumors only biopsy-confirmed renal cancers were included. During follow-up, 46 cases of in- 18 complete ablations were found. There was a significant correlation between tumor size and the 19 ablation success rate. The success rate was 73.5%, 87.6% for lesions ≤25mm, 94.6% for lesions 20 ≤25mm and exophytic, 79.1% for lesions 26-30mm, and 84.4% for lesions 26-30mm and exophytic. Three Clavien-Dindo grade ≥3 complications were observed. US-guided percutaneous RFA of T1a 22 renal cancers is safe and well-tolerated. Its effectiveness depends on tumor size, with the best results for exophytic lesions smaller than 3cm. Most recurrent or residual tumors can successfully retreat with US-guided percutaneous RFA.
Article is well written, and will add positive to the literature. Add the following articles:
https://doi.org/10.3390/sym14112450
10.3238/arztebl.2015.0412
10.1080/02656736.2019.1647352
10.2214/AJR.12.10210
10.2214/AJR.20.23803
This article can be accepted.
Reviewer 3 Report
The paper is interesting and may offer some interesting observations on the debate on NSS in complex cases. The paper has some points of strength but major revision is required.
First of all, be consistent with the ablative treatment reported. In the Abstract you talk about RFA (radiofrequency ablation) while in the introduction you talk about thermal ablation in general (which means also cryo). Please be consistent and more precise.
Introduction: When presenting the thermal ablation treatment, please cite guideline indications such as 3 cm limit / preferred for unfit and comorbid patients.
When talking about the difference between the surgery approach vs ablative please consider also:
- Lower Hb drop
- Shorter OT
For this scope please up-to-date the sentence in lines 36-37-36 of the introduction by citing the following recent paper on the matter: DOI: 10.1089/end.2022.0478;
Introduction: When describing renal nephrometry score please cite also the PADUA which remains one of the most used worldwide
The treatment failure is confirmed at the first follow-up, when do you perform this visit? How many months after treatment? Please define the cohort precisely given the big occupied slice in the result.
Single kidney
Discussion: “Some authors reported comparable results of thermal ablation and NSS in selected patients”. Please consider including more recent and strong evidence as suggested. The same is for the complication rate. Considering the challenging scenario of the completely endophytic renal masses, a recent multi-institutional showed a lower overall and minor complication rate for the ablative treatment compared to RAPN. For this scope please cite those interesting papers on the topic:
- Single kidney patients. In this setting recent evidence showed how PTA can be safely offered. In general, it should be preferred in more frail patients to minimize the risk of complications. Compared with RAPN, it might carry a higher risk of recurrence; on the other hand, re-treatment is possible (DOI: 10.1016/j.ejso.2022.09.022).
- Completely endophytic renal masses: (DOI: 10.1089/end.2022.0478; already suggested before) in this setting PTA confirms, again, to be an effective treatment for completely endophytic renal masses, offering low complications and good mid-term functional and oncologic outcomes. These outcomes compare favorably with those of RAPN.
Check typos
Round 2
Reviewer 3 Report
The authors should be congratulated for presenting an improved revised version of the manuscript. In my opinion, the manuscript is worthy of publication.